# Chemodiversity and Biotechnological Potential of Microginins

**DOI:** 10.3390/ijms26136117

**Published:** 2025-06-25

**Authors:** Joaquim da Silva Pinto Neto, Gustavo Marques Serra, Luciana Pereira Xavier, Agenor Valadares Santos

**Affiliations:** Laboratory of Biotechnology of Enzymes and Biotransformation, Biological Sciences Institute, Federal University of Pará, Belém 66075-110, PA, Brazil; gustmarq4@gmail.com (G.M.S.); luxavier@gmail.com (L.P.X.); valadaresantos@gmail.com (A.V.S.)

**Keywords:** cyanobacteria, peptide, microginin, chemodiversity, biotechnology

## Abstract

Biotechnology has increasingly focused on cyanobacteria as these microorganisms are a rich source of secondary metabolites with significant potential for various industries. Cyanobacterial metabolites have been described to have a wide range of biological activities, including cytotoxicity in cancer cells, inhibition of pathogenic bacteria and fungi, and inhibition of various enzymes, demonstrating a great diversity of bioactive compounds. The cyanobacterium *Microcystis aeruginosa* is well known for its toxicity and production of the cyanotoxin microcystin. However, another peptide produced by this cyanobacterium, microginins, has significant biotechnological potential. These linear pentapeptides were initially discovered for their angiotensin-converting enzyme (ACE) inhibitory activity. Subsequent studies have explored the full potential of this peptide, revealing its ability to inhibit other enzymes as well. This review aims to compile and systematize the microginins with biotechnological potential described in the literature, as well as outline their main structural characteristics and the predominant methodologies for their isolation and identification.

## 1. Introduction

Cyanobacteria are microorganisms of great environmental and biotechnological importance, as they were among the first living beings on Earth, appearing more than 3.5 billion years ago, and were largely responsible for oxygenating the Earth’s atmosphere [1,2,3]. Over the years, these microorganisms have become the focus of studies on their ability to produce molecules with diverse biological activities. Since the 1990s, several research groups have turned their attention to these organisms to provide new sources of bioactives for industries, that are constantly interested in formulating new products more quickly, profitably, and with better quality than those already available on the market.

These organisms stand as crucial reservoirs of bioactive natural substances, renowned for their synthesis of secondary metabolites, proteins, and enzymes capable of exerting cytotoxic, antifungal, antibacterial, and antiviral effects [4]. The bioactive compounds from cyanobacteria can be used by various industries, such as cosmetics, food, pharmaceuticals, and cleaning products, due to their diverse biological activities and functions. The production of bioactive molecules by these organisms is crucial to ensure the defense and survival of species, serving as an evolutionary mechanism that has provided adaptive advantages for cyanobacteria, enabling them to adapt to a wide range of habitats and protect themselves against external attacks from other cyanobacteria, microorganisms in general, and predators.

Different genera and species of cyanobacteria are described in the literature as major producers of bioactive molecules. Cyanobacteria of the genus *Microcystis* have great potential to produce bioactive compounds, but they are best known for their toxic and dangerous blooms, which release cyanotoxins into the environment, with microcystins being the widely study of these [5]. These freshwater cyanobacteria, especially *Microcystis aeruginosa*, are well known for producing microcystin, a non-ribosomal peptide that has been widely studied due to its ability to inhibit certain enzymes and possess cytotoxic activity [6]. Additionally, it was in this species that the production of a new peptide, microginin, was first described [7].

In this study, Okino et al. (1993) [7] discovered a new molecule with the ability to inhibit the angiotensin-converting enzyme (ACE). This new molecule was characterized as a linear pentapeptide and named microginin, due to its origin from the freshwater cyanobacterium *M. aeruginosa*, from the NIES collection (Microbial Culture Collection, the National Institute for Environmental Studies, Japan). The full benefits and mode of action of this peptide are still not fully known [8]

Microginin is a type of linear pentapeptide that stands out due to the presence of the Ahda group (3-amino-2-hydroxydecanoic acid) and the predominance of two tyrosine units at the C-terminus [7]. These peptides are mainly known for being ACE inhibitors, which underscores their biotechnological value as these types of inhibitors are used for the treatment of hypertension and heart failure [9]. In addition to this bioactivity, microginins have been identified as inhibitors of aminoproteinases (AMP), specifically aminopeptidases M, N, and leucine aminopeptidase, and they exhibit eco-cytotoxic activity against some crustaceans, nematodes, and fish in the larval stage [10]. Recent studies have indicated the possibility of using these peptides for cancer treatment due to their cytotoxic activity [11,12]. To date, the group of microginins contains more than 80 secondary metabolites from cyanobacteria [13]. In this review, we identified 123 distinct and analogous structures of microginins.

## 2. Structural Diversity of Microginins

The class of microginins is recognized as linear lipopeptides with the main characteristic of having a decanoic acid derivative, 3-amino-2-hydroxydecanoic acid (Ahda) [14] and the predominance of tyrosine in the C-terminal (Figure 1). These vary in length from 4 amino acids (e.g., microginin AL584, 91-A and -B) [15,16], to 6 amino acids (e.g., microginin 299-A and -B). Position 2 is the most variable with 8 different amino acids reported in the literature. Usually, the last position of the peptides can contain proline, tryptophan, or tyrosine, the latter being the most abundant in microginins.

N-methylation can occur at positions 1 to 5 [10,17,18]. Aliphatic mono-chlorination for Ahda was first reported by Ishida et al. (1997) [17] and, in some cases, N-terminal dichlorination may be present [16]. The mass range of microginins goes from 511.68 Da for microginin 511 (Ahda-MeMet-Tyr) to 931.12 Da for microginin 51-B (Ahda-Tyr-MeVal-MeTyr-Pro-Tyr) [16,18].

As mentioned in the introduction, Microginins were first reported and described in the literature from the toxic cyanobacterium *M. aeruginosa* (NIES-100) [7], which is well recognized for producing hepatotoxic peptides, the microcystins [19]. Microginin amino acid sequence was deduced by HMBC and ROESY correlations: Ahda-Ala-Val-MeTyr-Tyr (Figure 1a). The molecular form of microginin was determined as C_37_H_55_N_5_O_9_ [7]. However, the stereochemical study of the C-3 Ahda of Microginin 1 was only elucidated a year later by Bunnage et al. (1995) [20], and it was determined that the natural occurrence of Ahda has *syn* relative stereochemistry and absolute stereochemistry (2*S*,3*R*) (Figure 2). This discovery was made by comparing naturally occurring spectroscopy data for Ahda against data for two synthetic homochiral stereoisomers, (2*R*,3*R*)-Ahda and (2*S*,3*R*)-Ahda.

In addition, the microginin isolated by Okino et al. (1993) [7] inhibited the activity of ACE with an IC_50_ of 7 µg·mL^−1^, however, there was no inhibition of other proteases such as papain, trypsin, chymotrypsin, and elastase at 100 µg·mL^−1^.

Following the discovery of Microginin 1 [7,21] identified new microginins produced by the freshwater cyanobacterium *M. aeruginosa* (NIES-299). In this work, the named Microginins 299-A and -B, which have inhibitory activity against leucine aminopeptidase, were isolated and their structure was elucidated. Microginin 299-A consists of six amino acids whose sequence is Cl-Ahda-Val-N-MeVal-N-MeTyr-Pro-Tyr and has a molecular mass (MW) of 887.50 Da. As for the constitution of microginin 299-B, it is closely related to microginin 299-A. They differ only in the residue that has a Cl_2_ Ahda, with a MW of 921.94 Da. The small difference between the structures of microginins 299-A and -B highlights the importance of their inhibitory activity against leucine aminopeptidase with IC_50_ of 4.6 and 6.5 µg·mL^−1^, respectively.

One year later, Ishida et al. (1998) [17] elucidated the structures of two new structures isolated from the cyanobacterium *M. aeruginosa* (NIES-299), microginins 299-C and -D. Further investigations showed the identification of a linear peptide as a microginin from the cyanobacterium *Oscillatoria agardhii* (NIES-610) and two new peptides, microginins 99-A and 99-B, which were isolated from the cyanobacterium *M. aeruginosa* (NIES-99). Microginin 299-C is a linear peptide like microginins 299-A and B, whose N-terminus is not halogenated. Its Ahda stereochemical configuration was determined to be 2*S*,3*S*. Unlike the other Microginin 299s mentioned above, Microginin 299-D is composed of five amino acids, where the Tyr found in the last position is absent in this peptide. The first structures discovered and reported in the literature so far are shown in Figure 3 and Appendix A, respectively.

In addition, the cyanobacterial culture *M. aeruginosa* (NIES-99) was isolated from a bloom in Lake Suwa [22]. The amino acid sequences for microginins 99-A and 99-B were determined with N-terminal ClAda and Cl_2_ Ada, respectively, and followed by the sequence Tyr-Leu-N-MeTyr-Pro. Microginins 299-C and 299-D inhibited leucine aminopeptidase with IC_50_ of 2.0 and 6.4 µg·mL^−1^, respectively, however, Microginins 99-A and 99-B were not able to inhibit at 100 µg·mL^−1^. None of the peptides were able to inhibit ACE, papain, trypsin, thrombin, plasmin, chymotrypsin, and elastase at 100 µg·mL^−1^.

Concurrently with the work of Ishida et al. (1997) [21], Lake Teganuma was the target of the discovery of two new microginin structures, microginin T1 and T2, from water bloom material. Their sequences were determined to be Ala-Pro-Tyr-Tyr, where the N-terminal Ahda is halogenated by chlorine and the structure of T2 is not [23]. T1 microginin was able to inhibit the ACE enzyme and leucine aminopeptidase with similar values to 299-C microginin, with IC_50_ of 5.0 and 2.0 µg·mL^−1^. In addition, microginin T2 obtained similar inhibitory activity against leucine aminopeptidase, but greater activity against ACE, with IC_50_ of 7.0 µg·mL^−1^.

*O. aghardii* (NIES-610) was discovered in 1997 as a source of two new structures analogous to microginins, named Oscillaginin (Figure 4) [24], which are tetrapeptides with the Ahda chlorinated in one structure and non-chlorinated in the other. Oscillaginin A has an MW of 617 Da and is determined by the amino acid sequence ClAhda-Ser-N-MeVal-Tyr. The spectral data determined that Oscillaginin B is closely related to the first compound, except for the Ahda ending which is not chlorinated. The same β-amino acid residue was found in Microginin 1 [7] and its configuration was determined as 2*S*,3*R* [20]. Like oscillaginin, nostoginins are examples of the linear peptide class (Figure 4). In the C-terminal position of nostoginins, the 3-amino-2-hydroxydecanoic acid (Ahda) residue present in microginins is replaced by 3-amino-2-hydroxyoctanoic acid (Ahoa) (e.g., Ahoa-Val-MeIle-MeTyr) [25]. The Nostoginin BN741 (MW 741 Da) isolated by Ploutno and Carmeli (2002) [25] inhibited bovine amino peptidase activity with an IC_90_ of 1.3 M.

Ishida et al. (2000) [16] isolated, for the first time, two microginins with the N-methylated Ahda amino acid. These oligopeptides were named Microginin 478, isolated from *M. aeruginosa* extract (NIES-478). Although they are similar in Ahda motif, they have different amino acid compositions. Where Microginin 478 compromises the sequence MeAhda-Val-MeVal-MeTyr-Tyr and has high inhibitory activity against ACE, with IC_50_ 10.0 µg·mL^−1^.

Advances in omics are contributing to an ever-deeper understanding of the various compounds produced and new variants with potential applications for the pharmaceutical industry. The study by Rounge et al. (2009) [26] was one of the first to use genomic approaches and analysis to investigate the cyanobacterium *Planktothrix rubescens* NIVA CYA 98. In this study, microginins called oscillaginins A and B were identified through metabolomic analysis, whose sequences consist of the amino acids Ser-MeVal-HTyr, displaying the chlorinated and non-chlorinated Ahda motif, respectively. Notably, the gene cluster responsible for the biosynthesis of these metabolites was not found to be inserted into a phosmid nor subsequently expressed.

Recently, Eusébio et al. (2022) [8] proposed that the production of microginins is carried out by enzymes encoded in the Biosynthetic Gene Cluster (BGC). In their work, the group detected the putative mic BGC present in the cyanobacterium *M. aeruginosa* LEGE 91341 encoding a homologue of dimetal carboxylate halogenase. The genomic analysis reported 12 new microginin structures, including mono- and di-chlorinated as well as non-halogenated. These new structures contain the less common Ahda fatty-acyl portion and a new combination of amino acids compared to other microginins. In addition, neither Hphe nor mPro had previously been found in this large group of cyanobacterial peptides.

## 3. Occurrence and Ecology

Regarding the diversity of microginins in cyanobacteria and other peptides, a common question arises as to how their occurrence is determined according to taxonomic and geographical distributions. Non-ribosomal peptide biosynthesis requires a significant part of the cell’s energy and nutritional resources. To date, the group of microginins has more than 80 secondary cyanobacterial metabolites [12], which have been detected in various genera and species of cyanobacteria. The microginins in synthesis correspond to more than 80 variants described in strains or field samples of the genus *Microcystis* [12,27].

Many new case studies on the production of microginins by cyanobacteria present in water and blooms have been demonstrated in the last decade (Table 1—Appendix A contains all the structures reported so far). Microginins have been found in bodies of water in South and North America, Africa, Asia, and Europe, and congeners are more frequent in lakes, rivers, and reservoirs in Japan and Greece [7,10,16,21]. Among the cyanobacteria that produce microginins, the genus *Microcystis* stands out [28].

The structure of these new linear non-ribosomal peptides is mainly marked due to the presence of Ahda amino acid in the N-terminal portion. In general, they have 4 to 6 amino acids in their composition [14]. However, in the work of Stewart et al. (2018) [18] it was shown that microginins 511 and 527 are composed of only 3 amino acids and position two is sulfonated and N-methylated, this being one of the rare cases in which the biosynthesis of these microginins occurs.

The occurrence of antimicrobial activity by microginin was observed in the work of Gesner-Apter et al. (2008) [15]. Microginin AL584 was isolated from *Microcystis* sp. blooms and consists of the amino acid sequence ClAhda-Ala-N-MeVal-N-MeTyr. The inhibitor inhibited the proteolytic activity of trypsin at a concentration of 45.0 µg·mL^−1^, although it did not inhibit chymotrypsin, papain, or bovine aminopeptidase, the inhibitor was able to induce an irreversible stop in the growth of *Saccharomyces cerevisiae*.

In 2020, Zervou et al. [29] carried out a study on blooms and strains isolated from Greek lakes that were analyzed by liquid chromatography coupled to a linear ion trap/triple quadrupole hybrid mass spectrometer (LC-qTRAP MS/MS). A wide range of microginins that occur in cyanobacterial blooms co-occur in their synthesis with microcystins and other cyanopeptides. The most frequently detected microginin structures in the Greek Lakes were Microginin FR1 (70%), Microginin T1 (52%), Microginin 565B (52%), Microginin T2 (43%), and Microginin 565A (43%). In addition, other new strains of cyanobacteria i.e., *Nostoc oryzae*, *Synechococcus* sp., *M. aeruginosa*, *M. viridis*, and five *Microcystis* sp. [29].

Aquatic organisms in real-life scenarios are exposed to mixtures of several bioactive compounds and interactions between the components can result in additive and synergistic antagonistic effects [12,30]. The co-occurrence of microginins is well described in the literature. Carvalho et al. (2008) [31] described the formation of blooms in a reservoir in Brazil with cyanobacteria of the genera *Microcystis* and *Sphaerocavum* as dominant species, producing microginins, microcystins, anabaenopeptins B and anabaenopeptins F. Therefore, the concomitant production of various bioactive compounds and the growth of cyanobacterial strains is common in the literature. During different growth phases, the content of microginins can vary [32]. Changes can occur according to the nutritional conditions of the environment in which they are immersed [33] (Carneiro et al., 2012).

Carneiro et al. (2012) [33] suggest that, like microcystins, microginins may have an ecological function that protects cyanobacteria against the suppression of competitors. In this work, two new structures of two new microginin congeners were proposed (MG756 and MG770) isolated from the Salto Grande reservoir, located in São Paulo, Brazil, and identified through genomic DNA data and PCR amplification. The result of these co-production analyses concluded that species that produce microcystins can produce microginins together. Previously, Rohrlack et al. (2001) [34] isolated 13 strains of *Microcystis* from Lake Wannsee in Berlin, Germany, in 1995, which were found to give microcystins or anabaenopeptins and microginins.

*Microcystis* colonies collected from two water reservoirs in the Czech Republic produced more than 90 different cyanopeptides, i.e., anabaenopeptins, aeruginosins, microginins, cyanopeptolins, microcystins, and others [35]. Due to the simultaneous or successive mass development of different species of cyanobacteria in the same body of water, different metabolites can be released into the water and influence the organisms that inhabit it [36]. In summary, research by Bownik et al. (2022) [12] has shown that cyanobacterial oligopeptides such as microginin-FR1, anabaenopeptin-A, microcystin-LR and the set of these co-occurring affect behavior and molecular points such as neurotransmitter activity, muscle structure, nuclear DNA structure and cytotoxicity in *Brachionus calyciflorus*.

**Table 1 ijms-26-06117-t001:** List of microginin occurrences.

N.	Mass (Da)	Microginin	Sequence of Amino Acid						Species	Isolation Source	Ref.
			1	2	3	4	5	6			
1	714.40	1	Ahda	Ala	Val	MeTyr	Tyr	-	*M. aeruginosa* (NIES-100)	Lake Suwa (Japan)	[7]
2	727.0	FR1	Ahda	Ala	MeLeu	Tyr	Tyr	-	*Microcystis* sp. water bloom	Lake Waltershofen (Germany)	[37]
3	887.50	299-A	ClAhda	Val	MeVal	MeTyr	Pro	Tyr	*M. aeruginosa* (NIES-299)	Lake Kasumigaura (Japan)	[21]
4	921.94	299-B	Cl_2_Ahda	Val	MeVal	MeTyr	Pro	Tyr	*M. aeruginosa* (NIES-299)	Lake Kasumigaura (Japan)	[21]
5	853.05	299-C	Ahda	Val	MeVal	MeTyr	Pro	Tyr	*M. aeruginosa* (NIES-299)	Lake Kasumigaura (Japan)	[17]
6	757.76	299-D	Cl_2_Ahda	Val	MeVal	MeTyr	Pro	-	*M. aeruginosa* (NIES-299)	Lake Kasumigaura (Japan)	[17]
7	772.37	99-A	ClAhda	Tyr	Leu	MeTyr	Pro	-	*M. aeruginosa* (NIES-99)	Lake Suwa (Japan)	[17]
8	806.81	99-B	Cl_2_Ahda	Tyr	Leu	MeTyr	Pro	-	*M. aeruginosa* (NIES-99)	Lake Suwa (Japan)	[17]
9	769.96	478	MeAhda	Val	MeVal	MeTyr	Tyr	-	*M. aeruginosa* (NIES-478)	Lake Kasumigaura (Japan)	[16]
10	917.10	51-A	Ahda	Tyr	MeVal	MeTyr	Pro	Tyr	*M. aeruginosa* TAC-51	Lake Suwa (Japan)	[16]
11	931.12	51-B	MeAhda	Tyr	MeVal	MeTyr	Pro	Tyr	*M. aeruginosa* TAC-51	Lake Suwa (Japan)	[16]
12	732.34	T1	ClAhda	Ala	Pro	Tyr	Tyr	-	*Microcystis* sp*. water bloom*	Lake Teganuma (Japan)	[23]
13	698.38	T2	Ahda	Ala	Pro	Tyr	Tyr	-	*Microcystis* sp. water bloom	Lake Teganuma (Japan)	[23]
14	585.30	AL584	ClAhda	Ala	MeVal	MeTyr	-	-	*Microcystis* sp. TAU-IL306	Water reservoir Kibbutz (Israel)	[15]
15	575.19	91-A	ClAhda	Ile	MeIle	Pro	-	-	*M. aeruginosa* (NIES-478)	Lake Kasumigaura (Japan)	[16]
16	609.63	91-B	Cl_2_Ahda	Ile	MeIle	Pro	-	-	*M. aeruginosa* (NIES-478)	Lake Kasumigaura (Japan)	[16]
17	703.92	91-C	Ahda	Ile	MeIle	Pro	Tyr	-	*M. aeruginosa* (NIES-478)	Lake Kasumigaura (Japan)	[16]
18	738.36	91-D	ClAhda	Ile	MeIle	Pro	Tyr	-	*M. aeruginosa* (NIES-478)	Lake Kasumigaura (Japan)	[16]
19	772.81	91-E	Cl_2_Ahda	Ile	MeIle	Pro	Tyr	-	*M. aeruginosa* (NIES-478)	Lake Kasumigaura (Japan)	[16]
20	774.34	773	Cl_2_Ada	Pro	Phe	Pro	Tyr	-	*M. aeruginosa* LEGE 91341	Lake Braças (Portugal)	[8]

## 4. Biosynthesis

Many of the natural products produced by cyanobacteria either have a peptide section in their structure and are synthesized using the non-ribosomal peptide synthase (NRPS) or polyketide synthase (PKS) enzyme systems [8,34]. NRPS enzymes incorporate a range of proteinogenic and non-proteinogenic amino acids that can then be modified by optional domains for the N-methylation, epimerization, or heteroacylation of amino acids in the peptide structure. Although atypical, β-amino acids and their α-hydroxy derivatives play key roles in natural products due to their biological activity, as in the case of microginins as ACE inhibitors [35].

Microginins are synthesized in a non-ribosomal manner by a large complex of multienzymes comprising the NRPS and PKS modules. The microginin biosynthetic gene cluster (mic BGC) is approximately 30 kb long and comprises 8 genes, micA-H (Figure 5). The mic BGC has been characterized in cyanobacterial strains, including *Microcystis* and *Planktothrix* [38]. Although the group of putative genes for microginin synthesis has been sequenced [39], only in the work of Eusébio et al. (2022) [8] was the methodology for assessing the presence of microginin genes in *Microcystis* evaluated and described. The relatively limited methods for detecting microginin are generally based on HPLC/MS. In addition, PCR gene analysis using the microginin synthetase gene cluster can be used for the detection of new microginin variants or microginin-like structures [26].

In 2009, Rounge et al. [26] proposed that the synthesis of microginins is a group of NRPS polyketide hybrid genes (micACDE). This is found in the genome and is responsible for the production of oscillaginins, a type of the microginin family. The presence of NRPS domains, whose specificities correspond to the amino acids found in oscillaginins, and the co-linearity between this adenylation and the amino acid sequence in these peptides formed the basis for the proposal [8,26].

However, this reported proposal with a single PKS module and three NRPS modules failed to elucidate and explain the origin of the Ahda portion, which is the initial portion in microginins. Kramer [39] reported a fatty acid AMP-ligase (FAAL) enzyme associated with the BGC mic in *M. aeruginosa* and proposed that it would activate and load octanoic acid onto an acyl carrier protein (ACP), subsequently interacting with the PKS module to generate the Ahda residue. In line with the observations of Eusébio et al., 2022 [8], the genome data of *P. prolifica* NIVA-CYA 98 was updated, showing additional genes encoding FAAL, ACP, a dimethyl carboxylate halogenase and a SAM-dependent methyltransferase. These genes were analyzed and their role during microginin synthesis was confirmed.

Biosynthesis occurs first with activation of the fatty acid through the action of the enzyme FAAL, which activates octanoic acid (micA), followed by binding and loading by the acyl carrier protein (ACP) (micB) as a thioster. Halogenation of the fatty acid can occur by catalysis of the enzyme Dimetal carboxylate halogenase (micC), as well as N-methylations of amino acids that occur from the SAM-dependent methyltransferase (micH). The next stage is a PKS elongation module (micD) which includes the activity of the enzymes ketoacylsynthetase (KS), acyl transferase (AT), and acyl carrier protein (ACP2). AT is responsible for recognizing malonyl-CoA, KS is responsible for the Claisen-type condensation of activated octanoic acid adenylate with malonyl-CoA and ACP2 is responsible for binding the resulting decanoic acid. Finally, aminotransferase (AMT) is responsible for forming the β-amination of decanoic acid. This forms β-amino-α-hydroxydecanoic acid (Ahda) [8,26,39]. Table 2 shows the genes involved in the biosynthesis of microginins.

Depending on the microginin, it may have two to six NRPS elongation modules (micE/micF) which comprise at least the following activities: condensation domain (C), adenylation domain (A), thiolation domain (T). The A domain is responsible for activating the carboxyl groups of the amino acids, the T is responsible for binding and transporting the activated intermediate, and the C is responsible for condensing the amino acids with the growing peptide chain. Finally, the thioesterase (TE) activity cleaves the product of the microginin complex [39].

## 5. Isolation and Characterization Methods

The methods for identifying and characterizing microginin peptides involve a combination of analytical, biochemical, and spectroscopic techniques. In the first study to identify this peptide, Okino et al. 1993 [7] used analytical techniques such as Heteronuclear Multiple Bond Correlation (HMBC), chemical degradation, Rapid Atom Bombardment High-Resolution Mass Spectrometry (ROESY), amino acid analysis, and chromatography techniques to determine the structure of this peptide. These methods made it possible to identify the composition of microginin, including a new amino acid called Ahda. At the end of the article, the authors conclude with the structure of this bioactive peptide and the discovery of a new amino acid Ahda.

In a study carried out by Kodani et al. in 1999 [23], techniques such as Electron Activation Ionization Mass Spectrometry (FABMS) were used to determine the molecular mass of the peptides, Nuclear Magnetic Resonance, and High-Performance Liquid Chromatography (HPLC) to identify and characterize the compounds. Analysis of the NMR spectra and determination of the amino acid composition were essential for elucidating the structure and properties of the peptides. These methods allow peptides to be identified and, above all, their biological activities to be understood. Basically, almost all studies use similar techniques to identify the structure of the peptide and try to characterize it.

Following a common methodological order for the identification and characterization of peptides, the sample collected must first undergo a purification process, mainly using HPLC and its variants, such as Reversed Phase HPLC [10,15,17,21]. In some articles, the order is the opposite or combined, although in most studies the next step is the use of a mass spectrometry technique, which aims to determine the mass and amino acid composition of the peptide. Techniques such as ROESY [7], FABMS [16,23], liquid chromatography-triple quadrupole tandem mass spectrometry—LC-QqQ-MS/MS [27] and liquid chromatography triple quadrupole/linear ion trap hybrid mass spectrometry—LC-qTRAP MS/MS [29] have been used to determine the peptide’s mass.

In the first studies on these peptides and the most recent ones, the techniques used to characterize and define the molecular structure of microginins are nuclear magnetic resonance (NMR) [7,21,23,40,41,42].

## 6. Biotechnological Potential

In the scientific literature, the predominant focus on microginins is primarily directed toward their therapeutic potential. Studies and research highlight the relevance of these bioactive compounds in the search for new therapies and innovative drugs, with an emphasis on the pharmacological properties of microginins, especially their ability to inhibit enzymes such as angiotensin-converting enzyme (ACE), aminopeptidases, and others proteases.

The main activity of the lipopeptide microginin has been to inhibit the ACE protein. Several articles focus on this activity and test the potential use of this pentapeptide as a possible treatment for diseases linked to this and other enzymes [21,23,37]. ACE inhibitors are known to help treat cardiovascular and kidney diseases, with the ability to cause vasodilation, reduce blood pressure, and help treat heart failure, diabetes mellitus, diabetic nephropathy and stimulate natriuresis [43,44]. The best-known inhibitors on the market are captopril and enalapril [43].

The ACE protein has the function of converting the enzyme angiotensin I into angiotensin II, a natural biological process. Angiotensin II is responsible for vasoconstriction, increasing blood pressure, stimulating the release of aldosterone, and promoting sodium reabsorption in the kidneys. By inhibiting the conversion of angiotensin I into angiotensin II, ACE inhibitors end up promoting vasodilation, reducing sodium and water retention, and consequently lowering blood pressure [45].

Microginins have shown promise in biotechnology by being able not only to inhibit ACE activity but also aminopeptidase M, aminopeptidase N, leucine aminopeptidase (LAP) and trypsin [40]. Aminopeptidases are a class of proteolytic enzymes that belong to the family of exopeptidases and are responsible for hydrolyzing peptide bonds, removing the terminal amino acids from peptides and proteins. These enzymes operate in a specific way, separating the amino acids located at the N-terminus of the protein or peptide molecules, releasing them individually [46,47]. These enzymes have aroused significant interest in the pharmaceutical industry due to their broad potential to be incorporated into innovative drugs and therapies aimed at aiding and treating various diseases. Their promising applications include controlling inflammation, slowing tumor growth, aiding in cancer treatment, as well as managing blood pressure and hypertension [46,48].

In the 1999 paper by Kodani et al., [23] the ability of a microginin to inhibit an aminopeptidase, LAP, was highlighted for the first time. In this article, five new peptides were described, including two new microginins that were tested against serine proteases (trypsin, plasmin, and chymotrypsin), ACE, leucine aminopeptidase and carboxypeptidase A, as can be seen in Table 3. Both peptides inhibited angiotensin-converting enzyme and leucine aminopeptidase, increasing their biotechnological potential. The inhibition of leucine aminopeptidase, like that of other aminopeptidases, can be used as a way of treating some types of cancer, diabetes, and infections [47]. Other microginins were highlighted as aminopeptidase inhibitors in the study by Ishida et al. (2000) [16], in which five new microginins were able to inhibit the enzyme aminopeptidase M and could modulate enzyme activity under various conditions [47].

Aminopeptidase M is a type of these enzymes that is particularly found in bacteria, where it plays diverse roles, including metabolizing amino acids and participating in protein degradation processes. This enzyme has been tested in several studies with microginin [16,41,49], and in Ferreira’s study [49], two microginins, 756 (IC_50_ = 3.26 ± 0.5 μM,) and 770 (IC_50_ = 1.20 ± 0.1 μM) had inhibitory activity against aminopeptidase M as good as the inhibitor amastatin (IC_50_ = 0.98 ± 0.1 μM) (Table 3). This and other enzymes in this family, such as aminopeptidase N, are involved in the appearance and growth of tumors. A 2023 study by Brian Hur et al. [50] correlated pregnancy and the appearance and growth of a tumor with the biological activity of two enzymes called Endoplasmic Reticulum Aminopeptidase (ERAP) 1 and 2, and the article discusses how these enzymes may be potential anticancer agents, promoting the activation of immune responses against malignant cancers [50].

Of the class of aminopeptidase enzymes, the one that draws the most attention in cancer studies is aminopeptidase N (APN), also known as CD13, with numerous articles showing that inhibiting this enzyme can help treat cancer and its metastases [51,52,53,54]. The APN is present in several cell biological processes, especially in angiogenesis, the process of forming new blood vessels from existing blood vessels. Studies show that APN is overexpressed in tumor growth since tumors often induce the formation of new blood vessels to nourish them [54]. The 2011 review by Wickstrom et al. [53] discussed the association of aminopeptidase N with cell proliferation, secretion, invasion, and angiogenesis, as well as signaling different ways to approach this enzyme, such as directly inhibiting the enzyme or using APN as a biomarker for cancer.

The amastatin peptide has the potential to inhibit the ACE and some aminopeptidases, that can be administered orally through different formulations [49,55]. As presented in Ferreira et al., 2019 [49] in Table 3, Microginin 770 had an inhibition effect similar to the peptide amastatin, thus it can be concluded that this peptide is qualified to be used in a drug, from the point of view of the necessary inhibitory concentration, in addition to presenting characteristics in common with commercial ACE inhibitors, such as captopril and enalapril [56]. Further tests should be carried out to find out whether this and other microginins are susceptible to becoming drugs.

The mode of action of microginins is different due to the structure of the peptide and the specificity of the enzyme since the amino acid sequence of the peptide structure is directly related to its enzyme inhibition capacity. In general, the mechanism of microginins is to inhibit serine proteases. Basically, the peptide binds to the catalytic site of these enzymes, making it impossible for them to access substrates, so the enzyme cannot function properly and ends up inhibiting its action. This blockage can be reversible or irreversible, depending on the affinity of the microginin for the enzyme and the strength of the bond between them. In comparison, in the work of Paiva et al. 2017 [56], molecular docking was performed and the result suggested that microginin 770 interacts in the same way and at the same site occupied by the drug captopril, causing the same effect as it (in silico). This process alters the normal metabolism of organisms, but if administered correctly, microginins can be used in drugs and bring health benefits.

**Table 3 ijms-26-06117-t003:** List of inhibitory activity of microginin.

Microginin	Organism	Source	Ecosystem	Target	Activity (IC_50_)	Ref.
1	*M. aeruginosa* (NIES-100).	NIES-collection (Japan)	Freshwater	Inhibits angiotensin-converting enzyme	7.0 µg·mL^−1^	[7]
FR1	*Microcystis* sp.	Lake Waltershofen/Freiburg (Germany)	Freshwater	Inhibits angiotensin-converting enzyme	1.6 × 10^−5^	[37]
299 A	*M. aeruginosa* (NIES-299).	Lake Kasumigaura (Japan)	Freshwater	Inhibit leucine aminopeptidase,	4.6 µg·mL^−1^	[21]
T1	Field sample	Lake Teganuma (Japan)	Freshwater bloom	Leucine aminopeptidase/angiotensin-converting enzyme;	2.0 µg·mL^−1^/5.0 µg·mL^−1^	[23]
T2	Field sample	Lake Teganuma (Japan)	Freshwater bloom	Leucine aminopeptidase/angiotensin-converting enzyme;	2.0 µg·mL^−1^/7.0 µg·mL^−1^	[23]
51-A	*M. aeruginosa* (TAC-51)	Japan	Freshwater	Inhibits aminopeptidase M	4.5 (4.9 µM)	[16]
GH787	*Microcystis* spp.	Fishpond (Kibbutz Giva’at Haim—Israel)	Freshwater bloom	Inhibits bovine aminopeptidase N	7.7 mM.	[57]
756	*Microcystis* sp. LTPNA08/09	Salto Grande Reservoir	Freshwater bloom	aminopeptidase M	3.26 ± 0.5 μM	[49]
770	*Microcystis* sp. LTPNA08/09	Salto Grande Reservoir	Freshwater bloom	aminopeptidase M	1.20 ± 0.1 μM	[49]

## 7. Conclusions

The investigation into cyanobacteria, particularly the elucidation of microginins, underscores the significant biotechnological prospects inherent in these microorganisms. The array of bioactive molecules presents in cyanobacteria, exemplified by microginins, demonstrates a wide spectrum of biological activities ranging from enzyme inhibition to cytotoxic effects. The meticulous methods used to isolate and identify microginins in scientific studies elucidate their intricate structural features and pharmacological properties. With notable implications in pharmaceutical research, particularly as angiotensin-converting enzyme (ACE) and aminopeptidase inhibitors, microginins offer a promising avenue for the development of innovative therapeutic modalities in diverse medical fields, ranging from treatments for cardiovascular disorders to oncological interventions. The ongoing exploration of cyanobacteria and their bioactive constituents not only enriches our understanding of these microorganisms but also heralds a compelling trajectory for biotechnological advancements and the discovery of novel pharmaceutical agents.

## Figures and Tables

**Figure 1 ijms-26-06117-f001:**
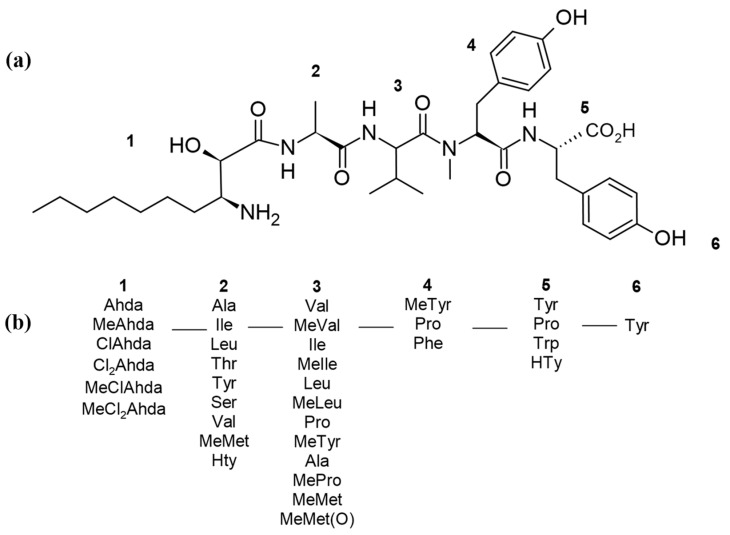
(**a**) First elucidated the structure of microginin, showing the numbers of the amino acid positions and the respective amino acids present in each position. (**b**) General scheme of microginins, illustrating the composition and organization of the positions and amino acids.

**Figure 2 ijms-26-06117-f002:**
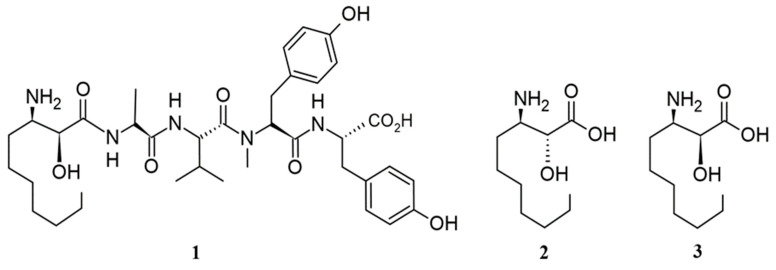
Ahda C-3 stereochemistry (1: solved stereochemical structure; 2: (2*R*,3*R*)-Ahda; 3: (2*S*,3*R*)-Ahda).

**Figure 3 ijms-26-06117-f003:**
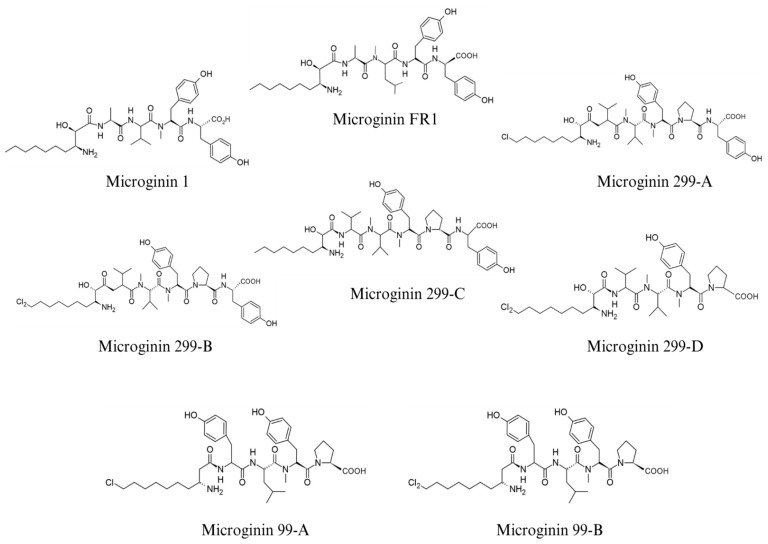
The structure of the first microginins is elucidated in the literature.

**Figure 4 ijms-26-06117-f004:**
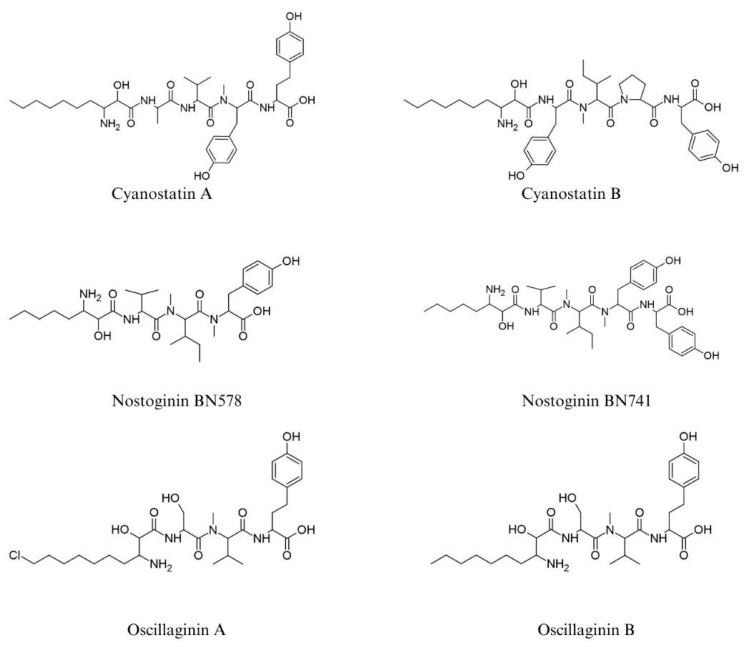
Structures analogous to microginins are reported in the literature.

**Figure 5 ijms-26-06117-f005:**
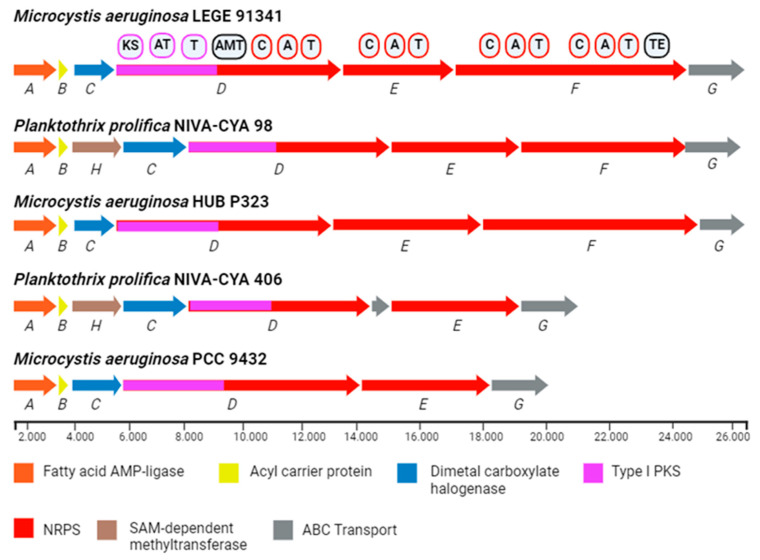
A graphical representation of microginin biosynthetic gene clusters. BGC: *M. aeruginosa* LEGE 91341 (accession BGC0002623); *Planktothrix prolifica* NIVA-CYA 98 (accession AM990468); *M. aeruginosa* HUB P323 (accession N.I.); *P. prolifica* NIVA-CYA 406 (accession DQ837301); *M. aeruginosa* PCC 9432 (accession CAIH01000183).

**Table 2 ijms-26-06117-t002:** Genes involved in the biosynthesis of microginins.

Gene	Products	Function
*mic*A	Fatty acid AMP-ligase	Activate the fatty acid as acyl adenylate.
*mic*B	Acyl carrier protein	Bind the fatty acid adenylate as a thioester.
*mic*C	Di-metal carboxylate halogenase	Halogenation of Ahda residues
*mic*D	Type I PKS/NRPS hybrid.	Elongation of the Ahda fragment and modification of the peptide
*mic*E	NRPS	Select and activate a specific amino acid for incorporation into the growing peptide chain.
*mic*F	NRPS	Selecting and activating a specific amino acid for incorporation into the growing peptide chain
*mic*G	ABC transporter	Transporting the peptide
*mic*H	SAM-dependent methyl transferase	Transfer methyl groups to the peptide.

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
