# Peer review of "Chemodiversity and Biotechnological Potential of Microginins"

_ijms, 2025, doi:10.3390/ijms26136117_

Round 1
Reviewer 1 Report
Comments and Suggestions for Authors
The review article on Chemodiversity and Biotechnological Potential of Microginins by Joaquim da Silva Pinto Neto et al., reviewed here the biotechnological potential class of linear lipopetide, Microginins. Although the selection of the compound class is interesting and worthy to review but I don't see much of novelty in the review article.
I find several repetitions of the information among different section of the article.
Also, I still think that this review article can be nicely organized with more proper information to come to publishable quality.
It is also recommended to include a section on Microginins and its potential biotechnological potential apart from just bioactivity. Also what are the novel approach to be applied for isolation of untapped source of novel Microginins.
I am attaching the file with my comments and suggestion.

Author Response
Response to Reviewer 1 Comments
Point 1: “I find several repetitions of the information among different section of the article”
Response 1: Thank you for this recommendation. We have already reviewed the article, and indeed, some information was repeated, which we have now corrected.
Point 2: “It is also recommended to include a section on Microginins and its potential biotechnological potential apart from just bioactivity. Also what is the novel approach to be applied for isolation of untapped sources of novel Microginins.”
Response 2: In the literature, we only find the biotechnological potential of microginin focused on its bioactive molecules, primarily on enzyme inhibition. Regarding its sources, microginins are peptides produced exclusively by cyanobacteria.
Point 3: “Corrections in the manuscript PDF.”
Response 3: As indicated in the manuscript PDF, the comments were accepted and reviewed. All scientific names were corrected and italicized. The information on the quantity of microginins was also updated. The captions for images 4 and 5 were modified based on your review. Regarding Figure 5, we can explain that to identify and build the biosynthetic clusters, articles describing the BGC were used and their access codes were obtained from them and analyzed in antiSMASH for a better understanding of the authors' own finalization in illustrating these BGCs.
Reviewer 2 Report
Comments and Suggestions for Authors
This review represents a compilation and systematization of microginins with biotechnological potential described in the literature: after introducing the historical evolution of the studies of microginins isolated from cyanobacteria, the authors outline their main structural characteristics and occurrence, and then review the biosynthetic aspects of this family of molecules and the predominant methodologies for isolation and identification; finally, always referring to the literature data, the potential biotechnological applications are illustrated and discussed.
I must say that the manuscript seems to be well arranged, clear, with distinct topics and well discussed.
I have just few suggestions:
1) I think that the paragraph "isolation and characterization methods" should be placed after the one concerning biosynthesis and before the one concerning biotechnological applications.
2) Reference 40 appears to be incomplete: are the authors referring to the patent "MICROGININ PRODUCING PROTEINS AND NUCLEIC ACIDS ENCODING A MICROGININ GENE CLUSTER AS WELL AS METHODS FOR CREATING MICROGININS" (WO/2007/062867), inventor Dan Kramer?
Author Response
Response to Reviewer 2 Comments
Point 1: “ I think that the paragraph "isolation and characterization methods" should be placed after the one concerning biosynthesis and before the one concerning biotechnological applications.”
Response 1: Thank you for pointing that out. We reviewed the article and decided to accept your recommendation and change the order of the topic.
Point 2: “Reference 40 appears to be incomplete: are the authors referring to the patent "MICROGININ PRODUCING PROTEINS AND NUCLEIC ACIDS ENCODING A MICROGININ GENE CLUSTER AS WELL AS METHODS FOR CREATING MICROGININS" (WO/2007/062867), inventor Dan Kramer?
Responde 2: Thank you for this correction. Indeed, that reference was incomplete. Due to the corrections made in the manuscript, the reference in question now has the number 39, instead of 40.
Reviewer 3 Report
Comments and Suggestions for Authors
It is an inetresting study which provides insight on biotechnological potential of Microginins. In this review, biotechnological potential of microginins, their main structural characteristics, methodologies for their isolation and identification have been provided. However, authors can address below comments to improve the manuscript:
· Is there downstream process involved in microginin production? Once microginins are produced by microorganism, how would they separate them from microbial cells. Include a section on downstream process if applicable.
· Provide a table on commercial use of microginin: companies procuring microginins and their specific application.
Author Response
Response to Reviewer 3 Comments
Point 1: “Is there downstream process involved in microginin production? Once microginins are produced by microorganism, how would they separate them from microbial cells. Include a section on downstream process if applicable..”
Response 1: Thank you for your review. The process of separation and purification of the peptide from cyanobacterial biomass has been presented in several articles and is indicated in the section "Isolation and Characterization Methods" on page 11 of our article.
Point 2: “Provide a table on commercial use of microginin: companies procuring microginins and their specific application.”
Responde 2: A very good idea to add to our article, but we currently do not have information in the academic literature about the production of microginins by companies. We have a table that highlights the biotechnological applications of some of these peptides (Table 3, page 14), which could potentially be utilized by industries in the future.
